# Generalizability of Total Worker Health^®^ Online Training for Young Workers

**DOI:** 10.3390/ijerph16040577

**Published:** 2019-02-16

**Authors:** Ashamsa Aryal, Megan Parish, Diane S. Rohlman

**Affiliations:** 1Department of Occupational and Environmental Health, University of Iowa, Iowa City, IA 52242, USA; diane-rohlman@uiowa.edu; 2Oregon Institute of Occupational Health Sciences, Oregon Health & Science University, Portland, OR 97239, USA; megan.parish@confluencehealth.org; 3Confluence Health, Wenatchee, WA 98801, USA

**Keywords:** young workers, training, Total Worker Health^®^, MTurk, health, safety, likeability, behavior change

## Abstract

Young workers (under 25-years-old) are at risk of workplace injuries due to inexperience, high-risk health behaviors, and a lack of knowledge about workplace hazards. Training based on Total Worker Health^®^ (TWH) principles can improve their knowledge of and ability to identify hazards associated with work organization and environment. In this study, we assessed changes to knowledge and behavior following an online safety and health training between two groups by collecting information on the demographic characteristics, knowledge, and self-reported behaviors of workplace health and safety at three different points in time. The participants’ age ranged from 15 to 24 years. Age adjusted results exhibited a significant increase in knowledge immediately after completing the training, although knowledge decreased in both groups in the follow-up. Amazon Marketplace Mechanical Turk (MTurk) participants demonstrated a greater increase in knowledge, with a significantly higher score compared to the baseline, indicating retention of knowledge three months after completing the training. The majority of participants in both groups reported that they liked the Promoting U through Safety and Health (PUSH) training for improving health and safety and that the training should be provided before starting a job. Participants also said that the training was interactive, informative and humorous. The participants reported that the PUSH training prepared them to identify and control hazards in their workplace and to communicate well with the supervisors and coworkers about their rights. Training programs based on TWH improves the safety, health and well-being of young workers.

## 1. Introduction

In 2016, there were approximately 19.3 million workers in the United States under the age of 24, representing 13% of the total workforce [1]. For 2016, incidence rates for non-fatal injuries and illnesses were 101.9 per 10,000 Full Time Employment (FTE) compared to 100.4 for all ages [2]. Similarly, in 2014, the rates of work-related injuries treated in emergency departments for workers, aged 15–19 and 20–24 were 2.18 times and 1.76 times greater than the rate for workers 25 years of age and older [3]. According to the Census of Fatal Occupational Injuries, the average rate of fatal injuries among workers less than 18 years was 47 deaths per year from 1994 to 2013 [4]. Inexperience, lack of knowledge about workplace hazards, and a reluctance to speak up have been associated with the increase in injury rates in young workers [5,6,7]. Young workers do not mention safety as their main priority at work and are often not aware of their legal rights and the tasks prohibited by labor laws [8]. They are eager to please their supervisors and may be reluctant to report injuries leading to underreporting [7,9,10], which might lead to an underestimation of the injury rates.

The Occupational Safety and Health Administration (OSHA) requires training to be a part of every employer’s safety and health program to protect workers from injury and illness (OSHA, 2015). Training programs have been found to improve knowledge and awareness of workplace safety [8,11]. However, most young workers report not receiving training on worker safety and health [12]. Those that receive training state that most trainings are brief and inadequate [13] and may not include information addressing both health and safety topics [14].

Recognizing these issues among young workers, the National Institute for Occupational Safety and Health (NIOSH) developed the Youth@Work: Talking Safety classroom-based curriculum to address the needs of young workers [15]. Promoting U through Safety and Health (PUSH), a Total Worker Health^®^ (TWH) training, expands the content of the Youth@Work curriculum to include information addressing health, safety, and communication in an online format. PUSH was developed through the Oregon Healthy Workforce Center, a NIOSH Total Worker Health^®^ Center of Excellence [12,14,16]. Total Worker Health^®^ is a strategy that integrates health promotion with injury prevention by looking at work as a social determinant of health. TWH focuses on job-related factors such as wages, hours of work, workload and stress levels, interactions with coworkers and supervisors, access to paid leave, and health promoting workplaces to have an impact on the wellbeing of workers [17]. Interventions addressing TWH improve workplace health effectively and more rapidly than wellness programs solely focused on health promotion [18,19,20].

Previously evaluated among parks and recreation workers, PUSH was found to be effective in increasing the safety and health knowledge among young workers [12]. The current study assessed the generalizability of the program among an expanded group of young workers. The main goal of the study was to assess the effectiveness of the PUSH training to increase knowledge about hazard identification, control selection, and communication between two groups of young workers using a pretest-posttest design. A second goal was to assess the likeability of the online training and to examine the impact on behavior to prepare workers to address hazards in the workplace (i.e. preparedness).

## 2. Materials and Methods 

### 2.1. Participants

The study was conducted in the United States in 2016. Two groups of young workers were recruited for the study: young workers employed at a city park and recreation program (Park and Rec) in the Pacific Northwest and young workers who were members of Amazon Marketplace Mechanical Turk (MTurk) who were located throughout the US. In order to be eligible for the study, the participants in both groups had to be less than 25 years of age and living in the United States. At baseline, 57 Parks and Rec workers agreed to participate after receiving a letter with details on the study during orientation. A second group of young workers were recruited online via Amazon Marketplace Mechanical Turk (MTurk): an open online marketplace for work that requires human intelligence [21]. Around 1000 MTurk workers answered a series of five questions as part of an eligibility screener. One hundred sixty-seven young workers between 18 and 24 years of age who met the eligibility criteria were recruited at baseline.

Park and Rec participants received $25 for completing the initial survey and training, and another $25 for completing the follow-up survey in the form of an Amazon gift card sent via email. MTurk workers received $0.02 to complete the initial screener, $20.00 to complete the training, and $4.00 to complete the follow-up survey through the MTurk platform. MTurk participants were paid less based on the time and difficulty of the task and because of the culture of the platform, where requestors (i.e., researchers) were discouraged from inflating payments to promote responses. The Oregon Health & Science University institutional review board approved the study materials and procedures.

### 2.2. Survey Instruments

Questionnaires were used to collect information at three time points: immediately prior to completing the training (Baseline), immediately after completing the training (Post-training), and 3-months after the training (Follow-up). Participants completed the questionnaire and the training online after they received a link to the materials either through their email or through the MTurk platform.

At baseline, participants provided demographic information along with information on work history and previous safety trainings. Health and safety knowledge was assessed immediately before and after the training by twenty-one multiple choice questions that were categorized into topics: hazard identification (e.g., “Sarah works at a bakery where her responsibilities are to take orders from customers, make sandwiches, and tidy up. Sometimes the morning rush is so overwhelming that she gets very distracted. What type of hazard is a distraction?”), control selection (e.g., “What is the least effective way of controlling a hazard?”), communication including workers’ rights, health behaviors, and safety questions, (e.g., “Regarding workplace violence, it is your responsibility to…”).

At the three-month follow-up, participants answered survey questionnaires about their current job, whether they liked the training and if they changed their behavior as a result of completing the training. Along with this, participants answered twenty-one knowledge questions and completed items addressing their general health and other health behaviors. Open-ended questions were used to assess the participants’ reaction to the training. Likeability and acceptability were assessed using the question "Did you like the PUSH training? Why or Why not?" and the impact of the training on behavior change was assessed through the question “Did you see or experience any behaviors over the last three months that you felt prepared to handle because of the PUSH training program? If so, please explain.”

### 2.3. PUSH Training 

The PUSH training is comprised of topics from the NIOSH Youth@Work: Talking Safety curriculum and two evidence-based curricula on health promotion [22,23], with additional topics addressing protection from workplace hazards, promotion of health and well-being, and workplace communication [12,16]. It was delivered through an online training format that had been used to teach skills using behavioral education principles among workers in different industries [14,24,25,26]. This format has also been effective in disseminating information on occupational health and safety for diverse worker groups [27,28,29].

Content experts in the field of occupational health and safety, and health promotion developed the original PUSH training. The videos and content used in the training were pilot tested with young workers on the MTurk platform as a part of the development process. Questionnaire items (e.g., demographics, work history, and likeability/acceptability) had been used previously by researchers in other studies with young workers [12,16]. The team used validated measures to assess general health, health behaviors, and job content [12,16].

PUSH is a self-paced online training that uses videos and real-life examples to teach young workers about safety, communication, and health. Participants were directed to a series of content screens with videos by an icon-based navigation system. Multiple choice questions were followed by brief videos that needed to be correctly answered to progress through the training [12]. Additional details in regard to the study are available at a separate study by the same co-author [12].

### 2.4. Statistical Analysis 

Data was analyzed using SAS 9.4 (SAS Institute, Cary, NC, USA). *t*-Tests and chi-square tests were used to examine differences in the demographic characteristics between the two groups at baseline. A mixed linear model with time-group interaction was used to evaluate the change in knowledge score at baseline, post-training and at follow-up. Responses to the twenty-one knowledge questions were marked “correct” and “incorrect”, denoted by 1 and 0, respectively, to create a cumulative score used to analyze change in knowledge among the participants. Due to the differences in age of the participants in the two groups, age was adjusted as a covariate in the model. To evaluate likeability and preparedness, responses to the open-ended questions were grouped into positive, neutral and negative categories for each group and were further coded to identify common themes.

## 3. Results

### 3.1. Demographics

We started the study with 219 participants who completed the demographic information. There were 118 participants at baseline and post-training and 70 participants at follow-up. Participants in the MTurk group were significantly older ($\bar{X}$ = 22 years, ranging from 19 to 24 years) compared to the participants in the Park and Rec group ($\bar{X}$ = 16 years, ranging from 15 to 19 years). The highest level of education in the MTurk group was a graduate degree and technical school in the Park and Rec group (Table 1). There were a greater number of female participants in the Park and Rec group. The majority of participants were Caucasian. The MTurk group had also been in the workforce longer than the participants in the Park and Rec group (1.9 vs 1.3 years, respectively). Park and Rec workers perceived their health to be better compared to the MTurk participants. The seasonal Park and Rec workers were employed at a single location and participated in regular safety meetings. Whereas, the MTurk workers were employed in a range of workplaces such as retail, food service, construction, health care, public utilities, manufacturing and agriculture with a variety of employers with and without regular safety meetings. Similar to the results in previous studies [5,30], young workers in both groups reported the need for safety training before starting a job: 95% in the MTurk group and 86% in the Park and Rec group.

### 3.2. Knowledge

Participants in the Park and Rec group had higher knowledge scores at baseline than the MTurk participants (Figure 1). The scores significantly increased immediately after the training for both groups (post-training). Compared to post-training, the scores significantly decreased at the 3-month follow-up. However, the scores at follow-up were still higher than the baseline scores for both groups. Even though the Park and Rec group started with higher knowledge scores at baseline, the MTurk group had higher scores post-training and at follow-up. However, the group difference was not significant at baseline and post-training (*p*-value at follow-up: <0.05). The participants in both groups scored the lowest on the questions addressing hazard identification and control selection at each time point. Cronbach’s alpha values measured at each time point for internal consistency of the questionnaires were 0.39, 0.55, 0.56 for baseline, post-training and follow-up, respectively.

### 3.3. Likeability and Preparedness

Two open-ended questions were used to assess the likeability of the training and whether or not the training led to changes in behavior or responses to situations in the participant’s workplace.

#### 3.3.1. Likeability of the Training

All participants completed the open-ended question about whether they liked the training. Most (63%) replied positively, while 15% answered neutral and 22% had a negative response. These responses were coded and grouped into categories that addressed the content of the training, the delivery format, or specific skills that were learned as a result of taking the training.

Content

Thirteen percent of the MTurk participants and thirteen percent of Park and Rec participants found the content of the training to be informative and stated that it provided useful information. For example, “*I liked the PUSH training because it was very informative in teaching the workers about looking out for those around them and also for their own wellness too.*” [MTurk, Restaurant Cashier] and “*Yes because it was informative and useful for future reference.*” [Park and Rec].

A few participants (8%) compared PUSH to other safety trainings they had taken in the past. The MTurk participants preferred the PUSH training, “*Yeah, I really liked it. Most job trainings are really boring, but the PUSH training was engaging. I remember enjoying it.*” [MTurk, Service manager]; whereas the participants in the Park and Rec group mentioned that they had learned most of the safety information in the PUSH training from their prior onsite training, “*I thought it had good intentions but I learned more about safety procedures from my on-site training at work.*” [Park and Rec worker].

Delivery format

The training is self-paced and divided into topics that include pictures and videos. Several participants (19%) felt the training was interactive and engaging and that they liked the training interface, “*I really enjoyed it. It presented information in an interesting, concise way. The time it took to take really went by quickly because the videos and interaction were so engaging.*” [MTurk, Childcare Provider] and “*I liked it because it was very interactive.*” [Park and Rec]. Participants’ comments identified the engaging/interactive nature (36%), the humor in the training (29%), and reported that it was easy to understand (21%), “*I did. I really enjoyed the way that it was formatted and felt as though it was helpful and funny without being corny or boring.*” [MTurk, Admissions counselor] and “*I enjoy the PUSH program because it’s both friendly and easy to understand.*” [MTurk, Office assistant].

Specific skills

Several of the participants (17%) mentioned that they liked the training because it increased their knowledge and provided specific skills for their job. For example, *“………….. I think it was useful, especially the parts about legal rights.*” [Park and Rec].

Negative and Mixed Findings

In contrast to the MTurk group, some participants in the Park and Rec group felt that the training was repetitive and boring and they had learned this information in previous trainings, “*I didn’t like it because it seemed repetitive to the safety training I underwent in order to get my job with PP&R.*” [Park and Rec] and “*It was fine, long but I understand why it was long*” [Park and Rec]. Other participants indicated mixed impressions of the training. They felt that although it was not useful to them, it could be useful to others, “*It was fine. I didn’t learn a whole lot, but I can imagine it being useful for others.*” [Park and Rec]

#### 3.3.2. Preparedness

Although the majority of participants did not indicate any change in behavior when asked how they felt prepared to handle real life issues after completing the PUSH training, about 30% did provide an example. These responses were coded into categories that described an increase in awareness of safety and health hazards or identified specific changes in behavior. Most of these responses (67%) came from participants in the MTurk group.

Increased awareness

Twenty percent of the participants mentioned that the PUSH training increased their awareness of the hazards in their workplace and were able to apply information from the training in certain situations, “*Yes, I am more aware of the dangers that I may face in the working area.*” [MTurk]; “*I experienced/noticed things that had been put in place to help keep us safe.*” [Park and Rec], and “*There were times where I reflected on the push training to help me in certain situations.*” [Park and Rec].

Behavior changes

Participants in the MTurk group provided more examples of behavior change than participants in the Park and Rec group. Many described a specific change in their behavior as a result of completing the training. For example, thinking about potential hazards prior to starting a task, “*I am able to think through tasks and situations more effectively before starting them. Anticipating potential danger is very important in my workplace.*” [MTurk] and “*I made sure to create a safer workplace for myself. I start by cleaning my work space more often by removing pins and empty plastic bags that may cause me to slip and injury myself.*” [MTurk]. Other participants indicated reporting or “speaking up” about workplace hazards, “*I noticed chemicals were not being stored correctly and I made sure to tell my boss about the issue and correct the problem.*” [MTurk, Customer service representative], “*I thought it was easier to talk to people I manage about safety and how to take care of themselves on the job.*” [MTurk], and “*I have asked for help in a few situations where I thought I may have gotten injured from carrying something I wasn’t meant to carry by myself such as large tables.*” [MTurk].

Several described specific instances of where they changed their behavior, “*I was able to handle hostile patients over the phone better to the PUSH training.*” [MTurk], “*There was a fire near our office that forced us to put an evacuation plan into action that was inspired by me after participating in the PUSH program.*” [MTurk, Office assistant], “*I felt I knew how to better assert myself towards healthful choices.*” [MTurk], “*Within the past 3 months, I realized that I was more cautious with my actions especially when I was working with other people.*” [MTurk], and “*I used all my protective equipment when cleaning up the pool.*” [Park and Rec]. 

## 4. Discussion

Approaches focusing on education and training have shown improvement in workplace safety [8,11]. As a result of completing the PUSH training, knowledge increased significantly from baseline to post-training for both groups. Although knowledge decreased at follow-up, it still remained elevated compared to baseline. The training was received positively by an overwhelming majority of participants in both groups, with most of the participants reporting the training to be interactive and informative and compared it with other training programs:

“*Yes, because it provided me with lots of insightful information that I did not learn on my job. I was able to know how to quickly respond to emergencies on the job after using the training program.*” [MTurk, Cashier].

Several participants in the Park and Rec group mentioned that the PUSH training was repetitive. This is likely due to the fact that participants in the Park and Rec group receive mandatory safety training before starting their job and have regular safety meetings throughout the season. These meetings address many of the topics presented in the PUSH training and could be the reason for these negative responses. On the other hand, MTurk participants were from diverse workplaces including restaurants and retail stores. Few of these participants reported receiving safety training. The majority of the MTurk participants liked the training and felt the training prepared them for workplace hazards.

The questions evaluating participants’ reaction to the training included multiple choice on a Likert scale with items as well as open-ended questions. All the participants entered a response to the question on likeability, and many gave specific reasons why they liked/did not like the training. The majority of participants answered the questions about preparedness and several (21%) provided specific examples. These responses stated that the training prepared them to handle workplace hazards by increasing awareness and led to specific changes in their behavior. One participant in the Park and Rec group indicated, “*I felt empowered to take action in my workplace environment when I saw something that violated my workplace rights or somebody else’s.*” [Park and Rec]. Raising awareness about working rights and building confidence in young workers to “speak up” about hazards is extremely important in promoting health and safety. It is not uncommon for participants to leave open-ended questions unanswered [31]. However, everyone in the study at follow-up provided their response to the open-ended question on how the PUSH training prepared them for behavior change with majority providing specific examples.

The goal of the current study was to assess the generalizability of the online training among different groups of young workers. The current study included young workers in a range of occupations, including cashiers, accountants, service managers, counselors, and lifeguards. The changes in knowledge in the current study replicated previous findings reported in parks and recreation workers [12]. This study provided additional feedback about the training including a generally positive response about the format of the training and the need for training for young workers. In addition, many participants provided examples in the open-ended questions describing situations where they felt empowered to speak up about safety hazards or specific changes to their behavior they made in their workplace.

One of the study’s strengths is that it is the first study to include two groups of young workers that were diverse in terms of their work experience. Young workers hired as summer employees at a city parks and recreation center were recruited along with a diverse group of workers selected via an announcement placed on Amazon Mechanical Turk. However, participants in both groups reported the need for training on health and safety. They liked the training and reported that the PUSH training prepared them to handle health and safety hazards at the workplace. The prospective nature of the study provides information on retention of information among young workers. Although knowledge scores at the three-month follow-up showed a decline from the immediate post-test, the scores were still greater than baseline for both groups with non-seasonal workers getting better scores compared to the seasonal workers. The decline in scores can indicate a need for frequent reminders or trainings on safety and health. Another strength of the PUSH training is the online format and interactive content. Young workers are familiar and comfortable with technology [11], which makes the online format of the training an appropriate dissemination technique for younger adults [16].

Studies have reported the need for training workers on the identification and control of workplace hazards [5,11,30]. A survey of Latino youths under the age of 21 working in construction found that the majority of participants reported that the training they received did not include information on controlling workplace hazards [13]. The results from the current study also emphasize the need to include topics on hazard identification and control selection as part of training for young workers, as evidenced by a higher number of hazard identification and control selection questions missed at each time point by participants in both groups.

One limitation of the study is that several participants were lost to follow up. Only participants who completed all three surveys were included in the analysis. However, the participants who were lost to follow-up had similar knowledge scores compared to the participants in the study at baseline and post-training. Another potential limitation of the study might be its generalizability outside of the US. Hence, there is a need for additional research to identify if the training can be utilized and to understand how young workers outside the US will receive it.

## 5. Conclusions

PUSH is an online training program utilizing a Total Worker Health^®^ approach to address occupational safety and health for younger workers [12]. These results suggest the usefulness of online-training to improve the safety, health and well-being of young workers, which prepares them to better prevent workplace hazards. Due to its inclusion of comprehensive topics on health and safety and its acceptance by young workers in diverse work environments, the PUSH training could be expanded to young workers in other industries to increase their awareness on workplace rights and responsibility, health communication in order to promote health and safety, and improve well-being.

## Figures and Tables

**Figure 1 ijerph-16-00577-f001:**
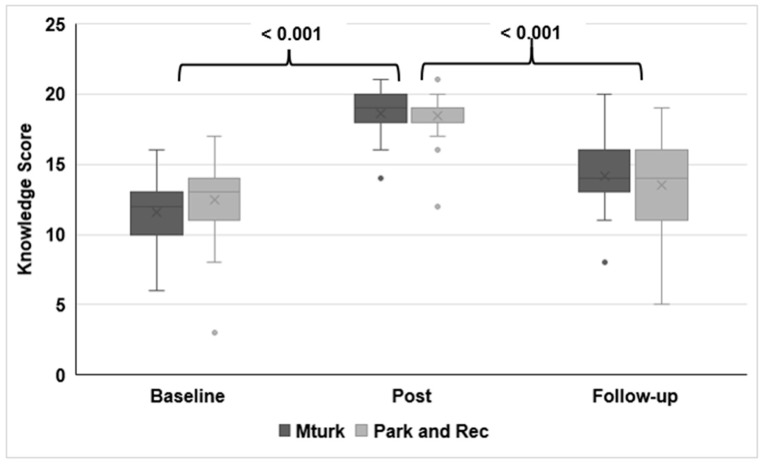
Box and Whisker plot showing change in knowledge score for the groups over time.

**Table 1 ijerph-16-00577-t001:** Demographic Data.

	Mturk (*N* = 39)	Park and Rec (*N* = 31)
	Mean (SD)	Mean (SD)
Age ***	22.4 (1.4)	16.03 (1.3)
Total years worked ***	1.9 (1.9)	1.3 (0.5)
	*n* (%)	*n* (%)
Gender		
Female	17 (43.6)	16 (51.7)
Ethnicity		
White/Caucasian	24 (61.5)	21 (67.6)
Asian/Pacific Islander	11 (28.2)	7 (22.6)
Others	4 (10.2)	3 (9.7)
Education ***		
High School	6 (15.4)	27 (87.1)
Technical school	15 (38.5)	4 (12.9)
College 4 years or more	6 (15.4)	0
College graduate or above	12 (30.8)	0

*** *p*-Value < 0.0001.

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
