# Peer review of "Generalizability of Total Worker Health® Online Training for Young Workers"

_ijerph, 2019, doi:10.3390/ijerph16040577_

Round 1

Reviewer 1 Report

Lines 47-48. 

However, most young workers report not receiving training on worker safety and health [11], those that receive training state that most trainings are brief and inadequate [12] and may not include information addressing both health and safety topics [13].

Recommendation (split into two sentences): 

However, most young workers report not receiving training on worker safety and health [11]. Those that receive training state that most trainings are brief and inadequate [12] and may not include information addressing both health and safety topics [13].

----

Lines 56-57. 

Perhaps a few more sentences explaining “Total Worker Health® approach” would be nice.  Given the information presented, this reviewer was not quite sure still as what this approach is.

----

Lines 70-71. 

Seasonal young workers less than 25 years of age, hired for summer at a 70 city parks and recreation center were eligible to participate in the study (Park and Rec).

Recommendation: 

Seasonal young workers less than 25 years of age, who were hired for summer at a city parks and recreation center, were eligible to participate in the study (Park and Rec).

----

Lines 113-115. 

Responses to the open-ended questions were grouped into positive, neutral and negative categories for each group and were further coded by to identify common themes.

Recommendation (eliminate “by”): 

Responses to the open-ended questions were grouped into positive, neutral and negative categories for each group and were further coded to identify common themes.

-----

Table 1

In Table 1, it is recommended to denote the total “n” for MTurk and Park and Rec students in the header.  Also it is assumed that the circles on Table 1's Box and Whisker plot are outliers.  If not, the circles on the plot need to be explained.

MTurk (n= )

Park and Rec (n= )

Lines 169-171

Several participants (19%) felt the training was interactive and engaging and that the liked the training interface

Recommendation (change “the” to “they”: 

Several participants (19%) felt the training was interactive and engaging and that they liked the training interface

-----

Line 173  The majority of the MTurk participants (26%) found the training to be humorous

Question:  How could the majority be considered 26%.

-----

Lines 215-221

Recommendation:  Place ALL specific comments from respondents in italics. Specific comments in these lines were not italicized.

-----

Lines 222- 224

This section may be divided by subheadings. It should provide a concise and precise description of the experimental results, their interpretation as well as the experimental conclusions 223 that can be drawn.

 Comment:  This statement seems to be comments on the manuscript that somehow got inserted into the manuscript.

 _____

Lines 258-259 

It is stated “However, everyone in the study at follow-up provided information on how the PUSH training prepared them for behavior change with majority providing specific examples.” 

 Question:  Isn’t this statement somewhat in conflict with previous statements made (lines 242-243? “Several participants in both groups provided specific examples stating that the training prepared them to handle workplace hazards by increasing awareness and behavior change. Although no specific behavior was identified

_____ 

Lines 276-277

A survey of Latino youths under the age of working in construction, found that the majority of participants reported that the training they received did not…

 Recommendation (eliminate comma): A survey of Latino youths under the age of working in construction found that the majority of participants reported that the training they received did not…

Author Response

Dear sir/madam,

Thank you very much for your constructive review on our manuscript titled, “Generalizability of a Total Worker Health® online training for young workers”. 

All your comments and suggestions have been addressed in the manuscript. The details on revisions with the responses are available in the file attached with the submission. 

Thank you again for your comments and suggestions. 

Best, 

Ashamsa 

Reviewer 2 Report

Thank you for asking me to review this manuscript.  

Overall, this is a well-written manuscript of  a study exploring the effectiveness of an online training for young workers to increase their knowledge on hazard identification, control selection and communication about health and safety, as well as to establish the program’s likeability and acceptance. I have some comments/query below: 

Section 1. Introduction

1.    Line 34-35: whilst this is factually correct, one is higher than the other, it is only marginally higher. I would state: “For 2016, incidence rates for non-fatal injuries and illnesses were 101.9 per 10,000 FTE compared to 100.4 for all other ages.” 

2.    Lines 36-38: These sentences are not really needed as this is older data and you have already established that this is a high risk group.  You might want to consider including recent data on fatal work-related injuries in this group. https://www.cdc.gov/niosh/docs/2017-168/pdfs/2017-168.pdf

3.    I would highlight next that this rate may in fact be lower due to under-reporting because of <lines 42-43>.

4.    An overall goal statement is included.  Is this the same as the overall purpose of this study?  What were the studies specific aims/research questions?

Section 2.  Material and Methods

1.    Please state the dates/year in which this study was conducted. Were all participants from the U.S.?  What state/region were the Parks and Rec participants from? 

Section 2.2: Survey Instruments

1.    Specifically state the eligibility criteria for both groups.  If there are any differences in the eligibility criteria for groups, explain the reasoning for this either here or in the discussion. 

2.    Please state whether any incentives and what these incentives were for both groups. How much were the MTurk participants paid?

3.    Can the questions included in each of the section be included in a table, or at least a couple of examples given for each category?

4.    An aim (as written in the goal statement) appeared to be to assess acceptance of the training.  How was this evaluated?

5.    How was the questionnaire/evaluation components developed? Was any testing done on the questionnaire prior to its use in this study?  How was content and construct validity established so we know that it is evaluating what you say it is evaluating?  Or did you use an already established and validated questionnaire?

6.    After reading the results section, I still feel like I am not entirely sure how you evaluated the program.  So, this really does need to be clearer and displayed more clearly e.g. table of both groups.

Section 2.3: Push Training

1.    I would like more information about the program/training: core curriculum/topics; more detailed information about the training methods. For eg., Was it content followed by questions? Or just content?  Were there any interactive elements with the training? 

2.    You show a graphic about the program – perhaps provide a little more information about what it is we are looking at with respect to the icons. It appears that this particular page has some content but also some reflective questions.  It also appears that there are 275 pages of training content. Is that correct?  Without any explanation, this graphic does not add much to the text.

3.    Was there any theory or evidence (e.g. from a systematic review or key studies) that informed the training content?  

4.    Who developed the content?  What did this process look like?  Was it piloted?  Did it undergo any testing prior to this study – e.g. cognitive testing of the content?

3.2 Knowledge

1. It appears you did some psychometric testing of the questionnaire itself (i.e. internal consistency)? But I am unsure what each value represents – is it at each time point, for each group at different timepoints? Also, this statistic seems out of place in this section.

3.3.2 Preparedness

1. This is a key section in the results but is not mentioned in the overall goal statement.  

2. Lines 222-224 seem to have been included in error?  

4.Discussion

1.  Please provide a statement on the generalizability of your findings.  Do you think this training would benefit all young workers? Is it U.S. specific (this is an international journal) or could it be applied to other countries?

2.  Please list any considerations that need to be made using an MTurk platform.  This is a different situation than a training that occurs for a job. How do you think this influenced how the participants engaged with the training and might have influenced the evaluation they completed? 

5.Conclusions

1. Please avoid claiming that this is the first online training program.  How do you know this for sure? 

Author Response

Dear sir/madam,

Thank you very much for your constructive review on our manuscript titled, “Generalizability of a Total Worker Health® online training for young workers”. 

All your comments and suggestions have been addressed in the manuscript. The details on revisions with the responses are available on the document (cover letter) attached with the submission. 

Thank you again for your comments and suggestions. 

Best, 

Ashamsa
